# An Examination of Multidimensional Time Perspective and Mental Health Outcomes

**DOI:** 10.3390/ijerph20064688

**Published:** 2023-03-07

**Authors:** Julia Moon, Rebecca J. Lieber, Ilke Bayazitli, Zena R. Mello

**Affiliations:** Psychology Department, College of Science & Engineering, San Francisco State University, San Francisco, CA 94132, USA

**Keywords:** time perspective, time feelings, time frequency, depressive symptoms, anxiety, rumination

## Abstract

We examined the associations between time perspective and mental health outcomes (*N* = 337; *M*_age_ = 22.74, *SD*_age_ = 5.59; 76% female). Time perspective included multiple dimensions (feelings, frequency, orientation, and relation) and time periods (past, present, and future). Mental health outcomes included depressive symptoms, anxiety, and rumination. To demonstrate the reliability of the time perspective scales, test–retest analyses were completed. Multivariate analyses showed that (a) positive feelings about time were associated with lower anxiety; (b) negative feelings about time were associated with greater anxiety; and (c) more frequent thoughts about the past were associated with greater depressive symptoms and anxiety. Associations remained when controlling for anxiety and depressive symptoms, respectively. Moreover, (a) positive feelings about time were associated with lower rumination; (b) negative feelings about time were associated with greater rumination; and (c) more frequent thoughts about the past were associated with greater rumination. Time perspective scales yielded scores that were moderate to high in test–retest reliability. Findings demonstrate the value of examining separate time perspective dimensions and time periods. Results highlight the role of time perspective in mental health interventions for adults.

## 1. Introduction

The prevalence of mental disorders and their implications for well-being highlight the critical need for identifying new ways to address mental health issues. About one in five adults have indicated mild to severe symptoms of depression, whereas one in six indicated symptoms of anxiety [1,2]. Depression, anxiety, and rumination have been positively associated with substance abuse [3,4], self-harm, and suicidal thoughts and behavior [5,6,7]. Further, the World Health Organization [8] has underscored the need for information about the distinct effects of specific mental disorders (i.e., anxiety separate from depression). Given the significance of mental health, it is imperative that this knowledge gap is addressed and that new targets of intervention for mental health be identified. Time perspective may be a powerful mechanism by which interventions can address poor mental health outcomes.

Time perspective broadly refers to feelings and thoughts about one’s past, present, and future [9,10]. Research on the construct has burgeoned with several conceptualizations emerging, including frameworks that focus on the past, hedonism, fatalism, and the future [10]; relationships among the time periods [11]; or whether the future is perceived as near or far [12]. In an effort to isolate the specific qualities of time perspective that are beneficial for intervention, Mello [9,13] proposed a model that comprises multiple distinct dimensions. This model builds upon previous work on future orientation [14,15,16] by including multiple time periods (past, present, and future). The model has been used in studies with adolescents, which indicate that time perspective dimensions are associated with mental health outcomes, including anxiety [17], stress [18], and self-esteem [19]. However, research has yet to use this model to examine mental health outcomes of adults. Further, the scales used to assess time perspective dimensions created by Mello and colleagues [9,13] have yet to be examined for test–retest reliability despite the charge by leaders in the fields of psychology and education to examine reliability in order to develop psychometrically robust scales [20]. Thus, the current study sought to (a) examine how time perspective dimensions were associated with mental health outcomes (depressive symptoms, anxiety, and rumination) among adults using a multidimensional model of time perspective [9,13] and (b) determine the test–retest reliability of the time perspective scales used [21].

### 1.1. Time Perspective

#### 1.1.1. Theoretical Conceptualization

Drawing from a long line of research that has conceptualized time perspective in a variety of ways [10,11,12], Mello and colleagues [9,13] developed a new model to advance our understanding of the construct. This model sought to generate distinct components of time perspective in order to understand which aspects were most strongly associated with human behavior and could be modified through intervention. These dimensions are organized into feelings and thoughts (frequency, orientation, and relation) about time (see Table 1). Time feelings encompass positive and negative feelings about each time period. A meta-analysis with adolescents and adults showed that both positive and negative time feelings were associated with poor outcomes including perceived stress, psychosocial maladjustment, and risky behavior [22]. The meta-analysis also indicated that on average individuals felt more positively than negatively about the past, present, and future. Additional studies with participants across the lifespan supported the six-factor structure of the construct (i.e., positive and negative feelings for each time period; [22,23,24,25]).

Time frequency refers to how often one thinks about the past, present, and future. Studies with young adults have demonstrated variations in the frequency of thoughts about time periods, with more frequent thoughts being reported about the present and future than about the past in general [23,26]. Time orientation is defined as the perceived relative importance of time periods. Prior research with young adults has shown that the most commonly reported time orientations are perceiving that (a) the future is more important than the past and present, (b) the present and future are more important than the past, and (c) all time periods are equally important [23,26,27,28]. Further, Konowalczyk et al. [26] showed that perceiving the present and future to be more important than the past was associated with greater optimism compared to perceiving just the present to be most important among young adults. Time relation refers to the perceived relationship among time periods. Research has indicated that most individuals across the lifespan perceive the present and future as related, all time periods as linearly related, or all time periods as interrelated [23,25].

#### 1.1.2. Mental Health Outcomes

Research has shown some associations between time perspective, depressive symptoms, and anxiety. For example, studies have indicated that greater levels of depressive symptoms are associated with less positive and more negative feelings about the past and present [29,30,31,32]. Studies have also shown that depressive symptoms are positively associated with having a lower emphasis on the future [33]. Regarding anxiety, studies have demonstrated that negative feelings about the past and future are positively associated with anxiety [34,35]. Åström et al. [36] have replicated these findings and also shown that less positive feelings about the past and future are reported in individuals with anxiety disorders compared to their counterparts. Findings from a meta-analysis have indicated that anxiety was negatively associated with placing a higher emphasis on the future [33]. Moreover, Finan et al. [17] have shown that adolescents who think that the past is the most important have greater anxiety than those who think that the present and future are more important than the past.

Importantly, knowledge is especially lacking with regard to the association between time perspective and rumination despite the call to generate knowledge about individual variations in rumination [7]. To our knowledge, there has only been one study that examined associations between any time perspective dimension and rumination [36]. This study included an adult sample and showed that negative feelings about the past and future were positively associated with rumination. Overall, a limited amount of research has examined associations between time perspective and mental health outcomes, and this research has included adolescents or focused on past and future time periods. We lack knowledge about these associations with adults and with all time periods.

#### 1.1.3. The Adolescent and Adult Time Inventory

The Adolescent and Adult Time Inventory (AATI) [21] was designed to assess multiple distinct time perspective dimensions (feelings, frequency, orientation, and relation). The AATI is increasingly becoming a common form to measure time perspective. Already, the AATI has been translated into more than a dozen languages (see author for translations) and included in research around the world. The psychometric properties that have been examined are promising but incomplete. For time feelings, the psychometric validity, internal consistency, and factor structure of the subscales have been extensively documented in the literature (for a review, see [22,37,38]). Findings from these studies have consistently supported the six-factor structure (i.e., six subscales that represent positive and negative feelings about the past, present, and future). However, the subscales’ test–retest reliability has yet to be examined, and this is an essential indicator of consistency [39]. The other remaining scales—including time frequency, time orientation, and time relation—are single items. The validity of these items has been demonstrated by several studies [19,26,40]. However, traditional psychometric analyses to determine structural validity and reliability are not possible. Thus, the current study sought to examine the test–retest reliability of the AATI measures [21].

Prior research with adult samples has demonstrated the test–retest reliability of measures assessing constructs similar to time perspective. For example, temporal focus—attention devoted to the past, present, or future—demonstrated moderate test–retest reliability over seven weeks [41]. Another measure assessing the tendency to think about immediate consequences was found to be moderately reliable over two weeks [42]. Lastly, a measure that examined perceived time left in life showed strong test–retest reliability over one year [43].

### 1.2. The Present Study

The present study included the following research questions: (a) How are time perspective dimensions associated with mental health outcomes (i.e., depressive symptoms, anxiety, and rumination)? and (b) Do the AATI measures of time perspective dimensions yield reliable scores across time? Time perspective dimensions were expected to be associated with depressive symptoms, anxiety, and rumination given prior studies (e.g., [30,36]). Specifically, positive time feelings would be negatively associated with these mental health outcomes, whereas negative time feelings would be positively associated. Moreover, thinking often about the past (time frequency), perceiving the future to be less important than other time periods (time orientation), and perceiving time periods as unrelated (time relation) were expected to be associated with greater levels of depressive symptoms, anxiety, and rumination. Further, the AATI measures were expected to yield scores that demonstrated test–retest reliability given research with measures assessing similar constructs (e.g., [41]).

## 2. Materials and Methods

### 2.1. Participants and Procedure

Convenience sampling was used to recruit participants at a public university in the western United States. Students attending a psychology course during the period of data collection were eligible. The institutional review board of the affiliated university approved the study procedures (Project Number 2020-050). Data collection occurred from the spring through the fall of 2020. The first survey (Time 1) was completed by participants on their own time on Qualtrics, a survey website. This survey assessed depressive symptoms, anxiety, and initial time perspective. The second survey (Time 2) was completed two weeks later and assessed rumination and follow-up time perspective. A two-week period is recommended between measurements to allow enough time to minimize carry-over effects while preventing actual changes in the measured construct [39]. Participants were offered course credit for participation at the discretion of their instructors. Data were analyzed with Stata (Version 14 and BE 17). Data and materials for the study are available by emailing the corresponding author.

The sample included participants who completed the first survey at Time 1 and comprised 337 individuals aged 18 to 72 years (*M*_age_ = 22.74, *SD*_age_ = 5.59). The sample age distribution in years was 18–28 (90%), 29–40 (9%), 41–52 (1%), 53–64 (0%), and 65–72 (<1%). Women (76%), men (21%), trans men (<1%), and non-binary/enby (3%) participated. The following racial/ethnic groups were reported by participants: African American/Black (8%), Asian American/Pacific Islander (23%), European American/White (17%), Hispanic/Latino(a) American (33%), multiple (15%), and other (5%). Maternal education was used as a proxy for socioeconomic status [44]. The sample maternal education ranged from 1 (no high school diploma/GED) to 6 (doctorate [MD/PhD]). The average maternal education (*M* = 2.60, *SD* = 1.26) was between a high school diploma/GED and an associate’s degree. No missing data were reported for key study variables.

A subsample was examined to assess the test–retest reliability of the AATI. The subsample included participants who completed surveys at Times 1 and 2 and comprised 178 individuals aged 18 to 52 years (*M*_age_ = 22.69, *SD*_age_ = 5.00). The subsample age distribution in years was 18–28 (90%), 29–40 (9%), and 41–52 (1%). Women (76%), men (21%), trans men (<1%), and non-binary/enby (2%) participated. The following racial/ethnic groups were reported: African American/Black (6%), Asian American/Pacific Islander (28%), European American/White (19%), Hispanic/Latino(a) American (31%), multiple (13%), and other (4%). The subsample maternal education ranged from 1 (no high school diploma/GED) to 6 (doctorate [MD/PhD]). The average maternal education (*M* = 2.73, *SD* = 1.29) was between a high school diploma/GED and an associate’s degree.

### 2.2. Measures

Time perspective dimensions were assessed with the AATI [21] at Times 1 and 2. Time feelings—positive and negative feelings about the past, present, and future—were measured separately with six five-item subscales (see Table 2 for alphas): Past Positive (“I have good memories about growing up”), Past Negative (“My past makes me sad”), Present Positive (“I am pleased with the present”), Present Negative (“I am not satisfied with my life right now”), Future Positive (“My future makes me happy”), and Future Negative (“I don’t like to think about my future”). Response options ranged from 1 (totally disagree) to 5 (totally agree). Subscales were generated by averaging items, with higher scores indicating greater positive or negative feelings. Previous research has demonstrated construct validity and a six-factor structure [37,38]. 

Time frequency—frequency of thoughts about the past, present, and future—was assessed with one item per time period (e.g., “How often do you think about your past?”). Response options ranged from 1 (almost never) to 5 (almost always). Prior research has used these items with adolescents [45] and young adults [26].

Time orientation was measured with a single-item categorical variable that included seven response options that displayed the time periods as circles (see Table 3, top). Participants were provided the following prompt: “Select one option below that shows how important the past, the present, and the future are to you, with larger circles being more important and smaller circles being less important.” Response options indicated the relative importance of the past (#1), present (#2), future (#3), past–future (#4), past–present (#5), present–future (#6), and all time periods (i.e., balanced; #7). 

Time relation was measured with a single-item categorical variable that included four response options (see Table 3, bottom). Participants were given the following prompt: “Select one option below that shows how you view the relationship among your past, present, and future, with touching circles being related to one another.”. Response options included unrelated (#1), present–future (only the present and future are related; #2), linearly related (#3), and interrelated (all time periods are related to each other; #4). Prior research has demonstrated the validity of the time orientation and time relation scales by showing that these variables are associated with mental health outcomes in adolescents [17,19]. Specifically, mental health is positively associated with response options that indicate orientation toward or interrelationships among multiple time periods. Further, studies with young adults have shown associations between time orientation and time relation with school type (preparatory or university; [23]).

Mental health outcomes were examined with the following measures. In the sample at Time 1, the 10-item Ruminative Responses Scale (RRS) [46] was used to measure rumination (e.g., “Go someplace alone to think about your feelings”; α = 0.87; *M* = 23.00, *SD* = 6.73). Importantly, this validated version of the 22-item RRS did not contain items that overlapped with depressive symptoms. Response options ranged from 1 (almost never) to 4 (almost always). Scores were generated by summing items, with higher scores indicating higher levels of rumination. In the subsample at Time 2, the 20-item Center for Epidemiologic Studies Depression Scale [47] was used to measure depressive symptoms from the past week (e.g., “I felt sad,” “I felt lonely”; α = 0.83; *M* = 42.07, *SD* = 10.61). Response options ranged from 1 (rarely or none of the time [less than 1 day]) to 4 (most or all of the time [5–7 days]). Scores were generated by summing items, with higher scores indicating higher levels of depressive symptoms. The 7-item GAD-7 [48] was used to measure symptoms of generalized anxiety disorder from the past two weeks (e.g., “Feeling nervous, anxious or on edge”; α = 0.93; *M* = 9.75, *SD* = 6.04). Response options ranged from 0 (not at all) to 3 (nearly every day). Scores were generated by summing items, with higher scores indicating higher levels of anxiety.

Age and gender were used as covariates. In the subsample at Time 2, life experiences (e.g., “Major change in financial status”; [49,50]) were measured and found not to be associated with time perspective dimensions in general, so this variable was not used as a covariate (see author for values). On average, participants reported experiencing about six life events with a positive impact (*SD* = 7.81) and ten with a negative impact (*SD* = 13.36) in the past four weeks.

### 2.3. Analytic Strategy

To examine the associations between time perspective and mental health outcomes (i.e., depressive symptoms, anxiety, and rumination), multiple linear regression and analyses of covariance (ANCOVA) were conducted. Covariates included age and gender. Depressive symptoms were also included as a covariate in models predicting anxiety, whereas anxiety was included as a covariate in models predicting depressive symptoms. Alpha adjustments were made for analyses with time feelings (six subscales; α < 0.008) and time frequency (three items; α < 0.017). Scheffé test was used for post-hoc comparisons. Effect sizes were interpreted according to Cohen’s [51] guidelines. 

To examine the test–retest reliability of the AATI, correlational analyses and chi-squared tests were conducted. Consistent with past research (e.g., [41,52]), we interpreted correlation coefficient values ranging from 0.50–0.70 to indicate moderate test–retest reliability and those above 0.70 to indicate strong test–retest reliability.

## 3. Results

### 3.1. Preliminary Analyses

Table 2 and Table 3 show descriptive statistics, and Table 4 presents correlations. Correlational analyses indicated that depressive symptoms, anxiety, and rumination had generally weak to strong negative associations with positive time feelings and positive associations with negative time feelings. These mental health outcomes also had weak to strong positive associations with thinking frequently about the past (i.e., past frequency).

Age was positively associated with positive thoughts about the present and negatively associated with negative thoughts about the present and future. Age was also associated with less frequent thoughts about the past and future. Further, age was negatively correlated with depressive symptoms, anxiety, and rumination. These effects ranged from small to moderate in size. Given these associations, age was controlled in all analyses (see Appendix A for estimates).

### 3.2. Time Perspective, Depressive Symptoms, Anxiety, and Rumination

For feelings about time (see Table 5, top), anxiety (after controlling for depressive symptoms) and rumination were negatively associated with positive time feelings and positively associated with negative time feelings. The effect sizes generally ranged from small to large. Analyses on depressive symptoms without controlling for anxiety showed that they were negatively associated with positive time feelings (present and future) and positively associated with negative time feelings (past and present) with small to medium effect sizes (see author for values).

For thoughts about time, thinking frequently about the past (i.e., past frequency) was positively associated with depressive symptoms and anxiety with small to medium effect sizes, after controlling for the other (see Table 5, bottom). Past frequency was also positively associated with rumination with a large effect size.

Time orientation was associated with rumination with a medium effect size ($\eta_{p}^{2}$ = 0.06; see Table 6, top). Although no significant pairwise comparisons were observed, descriptively, adults who indicated that (a) the past was more important than the present and future or (b) the past and future were more important than the present reported the greatest rumination compared to their counterparts. Analyses on depressive symptoms and anxiety without controlling for the other showed that they were associated with time orientation with medium to large effect sizes, but there were no significant pairwise comparisons (see author for values).

Time relation was associated with rumination with a small effect size ($\eta_{p}^{2}$ = 0.03; see Table 6, bottom). Descriptively, adults who perceived the past to be related to (a) the present or (b) the present and future reported the greatest rumination compared to their counterparts. Conversely, adults who perceived the past to be unrelated to the present and future reported the least rumination compared to their counterparts. Covariate effects were observed in all models (for the full models, see Appendix A). 

### 3.3. The Test–Retest Reliability of the Adolescent and Adult Time Inventory

Correlational analyses showed that the time feelings subscales had test–retest reliability coefficients ranging from 0.77 to 0.85 (*p*s < 0.001; see Table 2, top). Time frequency items (past, present, and future) had test–retest reliability coefficients ranging from 0.40 to 0.60 (*p*s < 0.001; see Table 2, bottom). For time orientation and time relation, chi-square tests showed that Time 1 and Time 2 responses were associated with large effect sizes (see Table 3).

## 4. Discussion

Many individuals suffer from poor mental health such as depression, anxiety, and rumination (e.g., [1,2]), all of which carry harmful implications for well-being (e.g., [3,6]). Examining time perspective and its associations with these mental health outcomes is important for efforts to use time perspective in interventions that prevent or treat mental health issues in adults. Toward this aim, it is critical to identify associations between time perspective and mental health outcomes that control for comorbid conditions. Further, given the wide scholarly interest in time perspective, there is a growing need to examine the test–retest reliability of the AATI, a measure for time perspective dimensions. Thus, the current study investigated (a) associations between time perspective dimensions (feelings, frequency, orientation, and relation) and mental health outcomes (depressive symptoms, anxiety, and rumination) and (b) the test–retest reliability of the AATI.

### 4.1. Time Perspective Dimensions Were Associated with Mental Health Outcomes

The present study provides several novel contributions to the literature. First, we used a multidimensional and multi-temporal model of time perspective to show its association with mental health outcomes in adults. Second, given the World Health Organization’s [8] recommendation to provide distinct information about depression and anxiety, we examined these associations while controlling for comorbid conditions. Third, we included rumination, a mental health outcome for which there is a dearth of information with regard to its association with time perspective. Importantly, although data were collected during the COVID-19 pandemic, we also measured life experiences (e.g., “Major change in financial status”; [49,50]), which were not shown to be associated with time perspective dimensions in general.

The current study extends prior research that focused on positive and negative feelings about the past and future only (e.g., [34,36]) by also showing that such patterns extended to the present. Specifically, findings indicated that feeling positively about the past, present, and future was negatively associated with anxiety, whereas feeling negatively about these time periods was positively associated with anxiety. The current study also yielded findings on relatively new dimensions of time perspective. For example, thinking frequently about the past was positively associated with depressive symptoms and anxiety. These results build on prior studies which demonstrated that emphasizing the future less was associated with greater depressive symptoms (e.g., [30]) by showing that emphasizing the past more was associated with greater depressive symptoms. Combined, these results show the value of examining multiple time perspective dimensions and time periods. 

We also conducted one of the first studies to examine time perspective and rumination. Our findings showed that adults who felt less positively about time periods, felt more negatively about time periods, or thought often about the past also reported more rumination than their counterparts. These results provide new knowledge and supplement available research which indicated that feelings about the past and future were associated with rumination [36]. We also observed that the relative importance placed on time periods and the perceived relationships among time periods were associated with rumination. However, the pairwise comparisons were not significant, most likely given the relatively small cell size per response option. These findings warrant further investigation and replication.

Our findings highlight the unique associations among time perspective dimensions, depressive symptoms, and anxiety. Further, the findings have implications for interventions utilizing time perspective. Specifically, such interventions may be especially effective by targeting specific time perspective dimensions based on whether individuals are experiencing symptoms of depression or anxiety. Prior intervention research has suggested that time perspective can promote positive physical and mental health outcomes. One intervention has demonstrated that thinking about the future consequences of present health-related decisions (e.g., going to the gym) resulted in greater thoughts about the future and physical activity than the control condition [53]. Another intervention that examined temporal constructs including nostalgia (past), gratitude for the current moment (present), and best possible self (future) showed that focusing on the present or future than the past resulted in a greater positive affect and sense of social connectedness during the COVID-19 pandemic lockdown [54]. Overall, our findings support the notion that time perspective may be a fruitful target for mental health interventions.

### 4.2. The Adolescent and Adult Time Inventory Demonstrated Test–Retest Reliability

Our findings addressed a gap in the literature and the call to consider measure reliability in the development of psychometrically robust scales [20] by showing that the AATI measures for time perspective dimensions had moderate to strong test–retest reliability. We examined the stability of scores across a two-week period based on guidelines [39]. The time feelings subscales and items for time orientation and time relation showed strong test–retest reliability. Time frequency items for the past, present, and future showed moderate test–retest reliability. These results are consistent with a prior study that reported similar test–retest reliability coefficients for time frequency as measured with a related construct [41]. Further, time frequency may be less stable in the current sample compared to other age groups, given that college students who are balancing short-term (e.g., examinations) and long-term (e.g., career preparation) goals may switch thinking about time periods more often. Future studies may conduct cognitive interviews [55] to examine how participants interpret the time frequency items. They may also investigate the association between time perspective and academic workload. Overall, the findings provide strong support for the use of the AATI measures to assess time perspective dimensions.

### 4.3. Limitations and Future Directions

Limitations of the current research include the cross-sectional research design. Future research should use a longitudinal study design to examine the directionality of the associations between time perspective and mental health outcomes. Another limitation is the generalizability of the college sample. Given that only two-thirds of high school graduates subsequently enroll in college [56], future studies may include samples from the general population in order to provide more generalizable findings. Moreover, genders were not equally represented in our sample given that there were more females than males. However, we do not believe that this substantively affected our results as prior research has not shown gender differences in time perspective [57] and gender was controlled in our analyses (see Appendix A for estimates). Additional research may examine unique associations between time perspective and mental health outcomes and the test–retest reliability of the AATI in other age periods, including adolescence or older adulthood. Further, given the high comorbidity of depression and anxiety [58], future research may investigate whether the observed patterns replicate in individuals with both mental disorders.

## 5. Conclusions

Associations between time perspective dimensions (feelings, frequency, orientation, and relation) and mental health outcomes (depressive symptoms, anxiety, and rumination) and the test–retest reliability of the AATI in a two-week period were examined among adults. Time perspective dimensions were associated with mental health outcomes. Further, the scores from the AATI measures assessing time perspective dimensions demonstrated test–retest reliability. These findings highlight the potential role of time perspective in mental health interventions and strongly support the use of the AATI to assess time perspective dimensions.

## Figures and Tables

**Table 1 ijerph-20-04688-t001:** Multidimensional time perspective model.

Category	Time Perspective Dimension	Definition	Sample Response
Feelings	Time feelings	Positive and negative feelings about the past, present, and future	“My past makes me sad,”“My future makes me happy”
Thoughts	Time frequency	How often one thinks about the past, present, and future	“Seldom,” “Almost always”
Time orientation	Perceived relative importance of the time periods	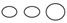 ^a^ (i.e., future is more important than the past and present)
Time relation	Perceived relationship among the time periods	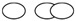 ^a^ (i.e., present and future are related but not to the past)

^a^ The circles represent the past, present, and future, respectively (for details, see Measures).

**Table 2 ijerph-20-04688-t002:** Descriptive statistics for time feelings and time frequency.

Time Perspective	Sample			Subsample						
				Time 1			Time 2			
	*M*	*SD*	α	*M*	*SD*	α	*M*	*SD*	α	*r* ^a^
**Time Feelings**										
Past Positive	3.14	0.79	0.86	3.14	0.79	0.88	3.20	0.81	0.91	0.85 ***
Past Negative	3.05	0.91	0.87	2.98	0.89	0.86	2.95	0.86	0.87	0.80 ***
Present Positive	3.42	0.75	0.91	3.35	0.77	0.92	3.40	0.79	0.93	0.77 ***
Present Negative	2.79	0.84	0.88	2.82	0.85	0.90	2.77	0.84	0.89	0.79 ***
Future Positive	3.96	0.76	0.92	3.91	0.79	0.92	3.85	0.81	0.93	0.85 ***
Future Negative	2.15	0.77	0.80	2.14	0.78	0.83	2.13	0.75	0.85	0.80 ***
**Time Frequency**										
Past Frequency	3.81	0.80	N/A	3.77	0.84	N/A	3.52	0.82	N/A	0.59 ***
Present Frequency	3.92	0.82	N/A	3.95	0.82	N/A	3.93	0.73	N/A	0.40 ***
Future Frequency	4.20	0.85	N/A	4.15	0.91	N/A	4.11	0.87	N/A	0.60 ***

The sample included participants who completed a survey at Time 1. The subsample included participants who completed a survey at Time 1 and the follow-up survey after two weeks at Time 2. ^a^ Test–retest reliability coefficients. *** *p* < 0.001.

**Table 3 ijerph-20-04688-t003:** Descriptive statistics for time orientation and time relation.

Time Perspective	Distribution, *n* (% ^a^)
Sample	Subsample
	Time 1	Time 2
**Time Orientation**				
1. Past	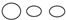	4 (1)	1 (1)	0 (0)
2. Present	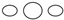	23 (7)	12 (7)	11 (6)
3. Future	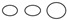	29 (9)	18 (10)	17 (10)
4. Past–Future	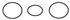	45 (13)	21 (12)	12 (7)
5. Past–Present	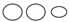	9 (3)	6 (3)	10 (6)
6. Present–Future	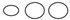	175 (52)	94 (53)	93 (52)
7. Balanced	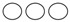	52 (15)	26 (15)	35 (20)
Pearson’s χ^2^ (*df*)		N/A	226.55 *** (30)
Cramér’s *V*		N/A	0.50
**Time Relation**				
1. Unrelated	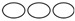	18 (5)	8 (4)	7 (4)
2. Present–Future Related	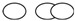	79 (23)	41 (23)	43 (24)
3. Linearly Related	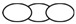	95 (28)	55 (31)	48 (27)
4. Interrelated	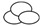	145 (43)	74 (42)	80 (45)
Pearson’s χ^2^ (*df*)		N/A	57.36 *** (9)
Cramér’s *V*		N/A	0.33

The sample included participants who completed a survey at Time 1. The subsample included participants who completed a survey at Time 1 and the follow-up survey after two weeks at Time 2. N/A = not applicable. ^a^ Column percentages may not add to 100 due to rounding. *** *p* < 0.001.

**Table 4 ijerph-20-04688-t004:** Correlations for time feelings, time frequency, and mental health outcomes.

Variable	1	2	3	4	5	6	7	8	9	10	11	12	13
**Time Feelings**													
1. Past Positive	—	−0.69 ***	0.22 ***	−0.19 ***	0.12 *	−0.09	−0.03	0.09	0.00	N/A	N/A	−0.26 ***	−0.01
2. Past Negative	−0.75 ***	—	−0.29 ***	0.38 ***	−0.15 **	0.27 ***	0.19 ***	−0.12 *	−0.01	N/A	N/A	0.40 ***	0.01
3. Present Positive	0.39 ***	−0.39 ***	—	−0.82 ***	0.44 ***	−0.43 ***	−0.12 *	0.17 **	0.06	N/A	N/A	−0.37 ***	0.13 *
4. Present Negative	−0.34 ***	0.47 ***	−0.83 ***	—	−0.45 ***	0.60 ***	0.24 ***	−0.11 *	−0.06	N/A	N/A	0.45 ***	−0.17 **
5. Future Positive	0.24 **	−0.26 ***	0.53 ***	−0.54 ***	—	−0.70 ***	−0.03	0.08	0.29 ***	N/A	N/A	−0.14 **	0.05
6. Future Negative	−0.20 **	0.35 ***	−0.45 ***	0.59 ***	−0.77 ***	—	0.13 *	−0.10	−0.19 ***	N/A	N/A	0.28 ***	−0.13 *
**Time Frequency**													
7. Past Frequency	−0.05	0.21 **	−0.19 *	0.22 **	−0.16 *	0.08	—	0.06	0.21 ***	N/A	N/A	0.44 ***	−0.21 ***
8. Present Frequency	0.09	−0.12	0.15	−0.12	0.14	−0.28 ***	0.08	—	0.12 *	N/A	N/A	−0.05	−0.05
9. Future Frequency	0.06	−0.06	0.13	−0.12	0.41 ***	−0.39 ***	0.10	0.28 ***	—	N/A	N/A	0.13 *	−0.19 ***
**Mental Health** **Outcomes ^a^**													
10. Depressive Symptoms	−0.15	0.20 **	−0.30 ***	0.29 ***	−0.20 **	0.18 *	0.44 ***	−0.03	0.05	—	N/A	N/A	N/A
11. Anxiety	−0.21 **	0.28 ***	−0.43 ***	0.46 ***	−0.26 ***	0.29 ***	0.39 ***	−0.09	0.09	0.60 ***	—	N/A	N/A
12. Rumination	N/A	N/A	N/A	N/A	N/A	N/A	N/A	N/A	N/A	N/A	N/A	—	−0.19 ***
13. Age	−0.07	−0.01	0.19 *	−0.19 *	0.08	−0.11	−0.11	0.10	−0.17 *	−0.17 *	−0.21 **	N/A	—

Correlations for the sample and subsample (data from Time 2) are shown above and below the diagonal, respectively. N/A = not available. ^a^ Depressive symptoms and anxiety were measured in the subsample at Time 2, whereas rumination was measured in the sample at Time 1. * *p* < 0.05. ** *p* < 0.01. *** *p* < 0.001.

**Table 5 ijerph-20-04688-t005:** Associations between time perspective (feelings and frequency) and mental health outcomes.

Time Perspective ^a^	Mental Health Outcomes ^b^														
	Depressive Symptoms					Anxiety					Rumination				
	*B*	*SE B*	ß	*F* ratio	$R_{\mathrm{adj}}^{2}$	*B*	*SE B*	ß	*F* ratio	$R_{\mathrm{adj}}^{2}$	*B*	*SE B*	ß	*F* ratio	$R_{\mathrm{adj}}^{2}$
**Time Feelings**															
Past Positive	−0.29	0.82	−0.02	24.84 ***	0.35	−1.05 *	0.44	−0.14	28.60 ***	0.38	−2.16 ***	0.44	−0.25	13.97 ***	0.10
Past Negative	0.54	0.78	0.04	24.98 ***	0.35	1.14 **	0.42	0.16	29.29 ***	0.39	2.93 ***	0.36	0.40	28.60 ***	0.19
Present Positive	−0.69	0.91	−0.05	25.02 ***	0.35	−1.95 ***	0.47	−0.25	33.37 ***	0.42	−3.14 ***	0.45	−0.35	22.83 ***	0.16
Present Negative	0.16	0.87	0.01	24.81 ***	0.35	2.15 ***	0.42	0.30	36.64 ***	0.45	3.45 ***	0.39	0.43	33.13 ***	0.22
Future Positive	−0.66	0.82	−0.05	25.05 ***	0.35	−1.07 *	0.45	−0.14	28.59 ***	0.38	−1.08 *	0.47	−0.12	7.50 ***	0.05
Future Negative	0.03	0.91	0.00	24.79 ***	0.35	1.57 **	0.48	0.20	30.61 ***	0.40	2.17 ***	0.46	0.25	13.40 ***	0.10
**Time Frequency**															
Past Frequency	3.15 ***	0.82	0.24	30.60 ***	0.40	1.06 *	0.49	0.14	28.24 ***	0.38	3.50 ***	0.42	0.42	30.18 ***	0.21
Present Frequency	0.43	0.89	0.03	24.89 ***	0.35	−0.54	0.50	−0.07	26.82 ***	0.37	−0.47	0.44	−0.06	6.07 ***	0.04
Future Frequency	−0.13	0.75	−0.01	24.80 ***	0.35	0.29	0.42	0.04	26.53 ***	0.37	0.81	0.43	0.10	6.93 ***	0.05

Age and gender were included as covariates in all models. Depressive symptoms were included as a covariate in models predicting anxiety, whereas anxiety was included as a covariate in models predicting depressive symptoms. Values for covariates are not shown (see Appendix A for the full models). ^a^ Alpha adjustments were made to account for the six models that examined time feelings (α < 0.008) and the three models that examined time frequency (α < 0.017). ^b^ Depressive symptoms and anxiety were measured in the subsample at Time 2, whereas rumination was measured in the sample at Time 1. * *p* < 0.05. ** *p* < 0.01. *** *p* < 0.001.

**Table 6 ijerph-20-04688-t006:** Associations between time perspective (orientation and relation) and mental health outcomes.

Time Perspective	Mental Health Outcomes ^a^
Depressive Symptoms	Anxiety	Rumination
*M* (*SE*)	95% CI	*M* (*SE*)	95% CI	*M* (*SE*)	95% CI
**Time Orientation**							
1. Past	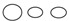	N/A	N/A	N/A	N/A	28.38 (3.50)	[21.49, 35.27]
2. Present	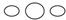	36.26 (3.42)	[29.48, 43.03]	9.41 (2.57)	[4.32, 14.51]	20.77 (1.46)	[17.90, 23.65]
3. Future	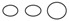	40.43 (2.32)	[35.84, 45.02]	12.15 (1.27)	[9.63, 14.67]	23.85 (1.23)	[21.44, 26.26]
4. Past–Future	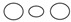	45.61 (2.75)	[40.16, 51.06]	8.25 (1.61)	[5.06, 11.45]	25.96 (0.98)	[24.02, 27.89]
5. Past–Present	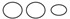	44.63 (3.24)	[38.21, 51.04]	10.76 (1.76)	[7.28, 14.24]	23.15 (2.17)	[18.88, 27.43]
6. Present–Future	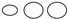	42.12 (0.96)	[40.21, 44.02]	9.11 (0.54)	[8.04, 10.19]	22.07 (0.49)	[21.09, 23.04]
7. Balanced	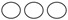	42.61 (1.57)	[39.50, 45.72]	10.59 (0.94)	[8.73, 12.45]	23.64 (0.92)	[21.82, 25.45]
*F* ratio (Radj2)		2.99 *** (0.36)	3.13 *** (0.45)	1.95 ** (0.09)
η^2^		0.55	0.66	0.18
**Time Relation**							
1. Unrelated	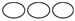	37.56 (3.60)	[30.44, 44.67]	12.64 (2.24)	[8.21, 17.08]	23.29 (1.56)	[20.21, 26.36]
2. Present–Future	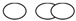	41.33 (1.47)	[38.43, 44.23]	9.34 (0.80)	[7.76, 10.92]	20.95 (0.75)	[19.48, 22.42]
3. Linearly Related	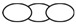	40.54 (1.37)	[37.82, 43.26]	11.03 (0.75)	[9.55, 12.51]	23.73 (0.68)	[22.39, 25.06]
4. Interrelated	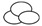	43.77 (1.03)	[41.72, 45.82]	8.94 (0.59)	[7.78, 10.11]	23.60 (0.55)	[22.51, 24.69]
*F* ratio (Radj2)		3.14 *** (0.37)	3.25 *** (0.46)	1.85 ** (0.07)
η^2^		0.55	0.66	0.16

Age and gender were included as covariates in all models. Depressive symptoms were included as a covariate in models predicting anxiety, whereas anxiety was included as a covariate in models predicting depressive symptoms. Values for covariates are not shown (see Appendix A for the full models). Adjusted values are displayed. No significant pairwise differences were observed. CI = confidence interval. N/A = not applicable because no participant selected this particular response option. ^a^ Depressive symptoms and anxiety were measured in the subsample at Time 2, whereas rumination was measured in the sample at Time 1. ** *p* < 0.01. *** *p* < 0.001.

## Data Availability

The dataset generated during and/or analyzed during the current study is available from the corresponding author upon reasonable request.

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
