# Peer review of "An Examination of Multidimensional Time Perspective and Mental Health Outcomes"

_ijerph, 2023, doi:10.3390/ijerph20064688_

Round 1

Reviewer 1 Report

The manuscript presents a number of  strengths including the multidimensional model of time perspective. The theoretical background is clear and concise. The results could be significant basis for the creation of certain psychotherapeutic interventions. The discussion section is also clearly written with reasonable explanations of obtained results. It would be useful to add differences in observed variables in different age groups.

  • The main question of the research is aimed to the study the associations between time perspective (researched through multiple variables) and mental health outcomes. 
  • The topic of the research is original and it is of relevance to the „International Journal of Environmental Research and Public Health“ readership. 
  • The similar research multidimensional model of time perspective has been used in adolescents, and this study focused on adult population. 
  • The methodology is clear and follows the research questions. It might be useful to show differences in the observed variables in different age groups (as the sample consists of participants between 18 and 52 years old in study 1, and 18 and 72 years old in study 2). 
  • The conclusion section presents the obtained results which were elaborated more specifically previously in the discussion section. 
  • The authors have used and cited new references that are significant for their research area. 

Reviewer 2 Report

Dear authors, thank you for the opportunity to get acquainted with your interesting research.

The relevance of the study is great. The problem of maintaining mental health is important for society. The article basically meets the requirements. The authors use reliable and valid methods. The reliability of the results is confirmed by mathematical statistics.

Despite this, I would like to make a few remarks.

1. It seems to me that the name should be changed, it is too pretentious. The authors study Time, but not Mind. It might be better to leave only An Examination of Multidimensional Time Perspective and Mental Health Outcomes.

2. I have questions about the sample description. Very large age range. "The sample included 178 individuals aged 18 to 52 years (Mage = 22.69, SDage = 5.00)"

Please specify how many people were in each age range.

3. The authors cite the data that "Studies have demonstrated individual variations in the frequency of thoughts about time periods, with more frequent thoughts being reported about the present and future than about the past in general [23, 24]". It is necessary to clarify the age of the subjects in these works. For older people, there are other data. They think about the past more often.

4. you need to improve the structure of the article.

- Authors do not need to describe the intention of their research in the Introduction (1.2. The Present Studies).

- It is not necessary to provide a selective description of the study procedure in the results (2.2.3.1.). This should be described in "2. Materials and Methods»

- The description of the research procedure is constantly repeated. It must be done 1 time ( 2. Materials and Methods)

- the authors poorly connected 2 Study in one article. Both studies should be combined into one and described as a single study. Both studies should first be described (2. Materials and Methods), and then their results should be presented.

Reviewer 3 Report

The concept of time perspective is incomprehensible to me, additionally tying this concept to the prediction of psychopathological symptoms of psychiatry seems a blind alley.

Reviewer 4 Report

The study came with results in support of multiple studies, namely that the individual's awareness of the past, present, and future is associated with psychological disorders. Despite this, the researchers followed the methodological steps to reach the results.

But later it was suggested to take into account the equivalence in the sample between females and males

Likewise, focusing only on one dimension of the time perspective, because its abundance causes confusion to the reader in the results

Round 2

Reviewer 2 Report

The authors have made the necessary corrections. I have no more comments